# Elevated Plasma Levels of Mature Brain-Derived Neurotrophic Factor in Major Depressive Disorder Patients with Higher Suicidal Ideation

**DOI:** 10.3390/brainsci13081223

**Published:** 2023-08-21

**Authors:** Haimei Li, Miaomiao Zhao, Chaonan Jiang, Haoyang Zhao, Congchong Wu, Ying Li, Shiyi Zhang, Pengfeng Xu, Tingting Mou, Yi Xu, Manli Huang

**Affiliations:** 1Department of Psychiatry, The First Affiliated Hospital, Zhejiang University School of Medicine, Hangzhou 310003, China; 11718207@zju.edu.cn (H.L.); miaomiaozhao@zju.edu.cn (M.Z.); chaonanjiang@zju.edu.cn (C.J.); haoyangzhao@zju.edu.cn (H.Z.); wucongchong@zju.edu.cn (C.W.); lucky_liying@163.com (Y.L.); zhangshiyi0118@163.com (S.Z.); xpfeng2022@163.com (P.X.); tingtingmou2015@126.com (T.M.); 1183030@zju.edu.cn (Y.X.); 2The Key Laboratory of Mental Disorder Management in Zhejiang Province, Hangzhou 310003, China; 3Zhejiang Engineering Center for Mathematical Mental Health, Hangzhou 310003, China

**Keywords:** BDNF-TrkB signaling, inflammatory factors, suicidal ideation, Beck Scale, major depressive disorder

## Abstract

Several pieces of evidence show that signaling via brain-derived neurotrophic factor (BDNF) and its receptor, tropomycin receptor kinase B (TrkB), as well as inflammation, play a crucial part in the pathophysiology of depression. The purpose of our study was to evaluate plasma levels of BDNF-TrkB signaling, which are inflammatory factors in major depressive disorder (MDD) patients, and assess their associations with clinical performance. This study recruited a total sample of 83 MDD patients and 93 healthy controls (CON). All the participants were tested with the Hamilton Depression Scale (HAMD), the Beck Scale for Suicide Ideation, and the NEO Five-Factor Inventory. The plasma level of selected BDNF-TrkB signaling components (mature BDNF (mBDNF), precursor BDNF (proBDNF), tyrosine kinase B (TrkB), and tissue plasminogen activator (tPA)) and selected inflammatory factors (interleukin-1β (IL-1β), interleukin-6 (IL-6), and tumor necrosis factor-α (TNF-α)) were measured using an enzyme-linked immunosorbent assay (ELISA). Further, we performed correlation analysis to indicate the relationship between the plasma levels of the factors and clinical characteristics. Results: (i) A higher level of mBDNF and lower openness were observed in MDD patients with higher suicidal ideation than patients with lower suicidal ideation. (ii) In MDD patients, mBDNF was positively correlated with the sum score of the Beck Scale for Suicide Ideation (BSS). (iii) The levels of mBDNF, tPA, IL-1 β and IL-6 were significantly higher in all MDD subjects compared to the healthy controls, while the levels of TrkB and proBDNF were lower in MDD subjects. Conclusion: Our study provides novel insights regarding the potential role of mBDNF in the neurobiology of the association between depression and suicidal ideation and, in particular, the relationship between BDNF-TrkB signaling, inflammatory factors, and clinical characteristics in MDD.

## 1. Introduction

Today, the total number of people suffering from major depressive disorder (MDD) is about to exceed 322 million globally [1]. The years of life lost due to disability (YLD) caused by depressive disorders has reached 43,099,000 (95% CI: 3,053,600–58,895,600), and the years of life lost due to depression have increased by 33.4% (95% CI: 31.0–35.8) compared to 1990 and by 14.3% (95% CI 13.1–15.6) compared to 2007 [2]. MDD is ranked by the World Health Organization (WHO) as the single largest contributor to global disability [3]. MDD as a syndrome has a range of symptoms and varies greatly in its course, symptomatology, and clinical outcome, leading to difficulties in treatment and mechanistic study [4]. In addition, the WHO has estimated that the overall international suicide rate is 10.6/100,000/year, accounting for 1.5% of all deaths [5]. However, there is no single model or mechanism that could explain all aspects of MDD satisfactorily, including suicidal behavior, attempts, or ideation.

Brain-derived neurotrophic factor (BDNF) is the most well-studied neurotrophin and has been implicated in the pathogenesis of various psychiatric disorders, including MDD. Both mature BDNF (mBDNF) and its precursor, proBDNF, play key parts in neuronal survival, morphogenesis, and plasticity. Via differential engagement with their receptors, they work in a largely opposed manner [6,7]. Despite this, proBDNF could be converted to mBDNF via intracellular and extracellular proteases, and the plasminogen tPA (tissue plasminogen activator)/plasmin system is the most crucial protease involved in this process [8]. mBDNF exerts biological effects on neurons by binding to its receptor tyrosine kinase B (TrkB), which subsequently elicits various intracellular signaling pathways, including mitogen-activated protein kinase/extracellular signal-regulated protein kinase (MAPK/ERK), causing oxidative stress and inflammation in depression [9,10]. proBDNF binds with high affinity to the p75 pan-neurotrophin receptor and is then implicated in depression and anxiety [11,12]. Multiple pieces of evidence indicate that BDNF metabolism is deleteriously altered in MDD patients as well as in animal models. However, the changes in the role of BDNF in depression are still elusive. Indeed, several studies have reported decreased BDNF levels in MDD patients [13,14], while others have demonstrated that BDNF levels were higher in MDD patients [15,16]. In addition, there have also been studies that have shown no significant difference in BDNF levels between MDD patients and controls [17]. One reason for these diverging results may be that most of the abovementioned research failed to differentiate mBDNF and proBDNF, or, due to the limitations of the kits available at the time, they might have included both forms of BDNF [18]. A recent study reported lower levels of serum and exosomal BDNF and higher levels of proBDNF in MDD subjects compared to controls, which indicates that mBDNF and proBDNF have different serum expression profiles [19]. According to the above, the expression of BDNF in MDD patients requires more elaborate analyses that include MDD patients with different characteristics.

Over the past years, there has been increasing evidence indicating that abnormalities in immune-inflammatory pathways and the activation of cell-mediated immunity represent important pathophysiological pathways that could contribute to the development of MDD [20,21,22]. Additionally, mounting experimental and clinical research has provided support for the psycho-neuro-inflammatory hypothesis. Several studies based on animals have found elevated concentrations of pro-inflammatory cytokines—IL-6, IL-1β, and TNF-α (interleukin-6, interleukin-1β, and tumor necrosis factor-α, respectively)—as well as anti-inflammatory cytokines in both the central and peripheral nervous systems in various rodents’ stress paradigms, namely the chronic mild stress (CMS) [23,24,25,26], social isolation [27,28], learned helplessness [29], olfactory bulbectomy (OB) [30], and social defeat [31,32] models of depression. Elsewhere, researchers have shown that a rapid antidepressant effect of ketamine is related to the downregulation of pro-inflammatory factors (IL-1β, IL-6, and TNF-α) in the hippocampus of rats [33]. Clinical studies have also found a significant association between numerous pro-inflammatory cytokines and depressive symptoms. MDD patients have been found to have increased concentrations of pro-inflammatory cytokines and acute-phase proteins, including IL-1, C-reactive protein (CRP), and monocyte chemoattractant protein-1, compared to non-depressed individuals [34,35,36]. Several meta-analyses have revealed elevated peripheral levels of IL-6 in MDD patients compared with controls, while the results for IL-1β and TNF-α differ among studies [37,38,39,40]. Taken together, the aforementioned results provide compelling evidence indicating the involvement of inflammatory factors in MDD.

However, there are currently no appropriate animal models to explain suicidal behavior. Hence, we started with MDD patients and the hypothesis that the expression levels of BDNF and immune factors in peripheral blood may be related to suicidal ideation in MDD patients. Additionally, the aim of the present study was to evaluate plasma levels of BDNF-TrkB signaling and inflammatory factors in medically stable, currently untreated MDD patients and assess their associations with clinical performance. We first suggested the correlation between mBDNF and suicidal ideation. The levels of pro-inflammatory factors (IL-1β, IL-6, and TNF-α) and factors involved in BDNF-TrkB signaling (mBDNF, proBDNF, TrkB, and tPA) in the peripheral blood of MDD and healthy controls were also examined to investigate the function of inflammation and BDNF signaling in MDD patients.

## 2. Materials and Methods

### 2.1. Subjects

In total, 83 unipolar MDD subjects were enrolled. All patients were admitted to the Department of Psychiatry, the First Affiliated Hospital, Zhejiang University School, for inpatient treatment. Their diagnoses were determined through the Structural Clinical Interview for the Diagnostic and Statistical Manual of Mental Disorders, fifth revision (DSM-V) and were confirmed by two expert psychiatrists. Healthy controls (CON) were recruited by clinical referrals, newspaper advertisement, and bulletin board notices. Control subjects had no history of any DSM-V axis disorder, which was also confirmed by clinical interview. All subjects had to have been free of any psychotropic medication for at least three months and exhibit a stable status. The demographics, including age, sex, education, and illness duration, are presented in Table 1.

Subjects were excluded if they: (1) had a current pregnancy; (2) had another current psychosis or a history of bipolar disorder or psychotic symptoms that existed outside a major depressive episode; (3) had an eating disorder or post-traumatic stress disorder (PTSD) within one month of entering the study; (4) presented alcohol or drug dependence or abuse within six months of entering the study; (5) had a history of neurological disorders, such as cerebral trauma, seizure disorder, or MRI evidence of structural brain abnormalities; (6) were MDD patients who had received any form of treatment prior to the study; (7) had medical illnesses that could be etiologically related to the ongoing depressive episode, such as acute infections, chronic inflammatory disorders, untreated hypothyroidism, uncontrolled hypertension, or diabetes; (8) refused to participate in the study. These criteria are also presented in Table 2.

### 2.2. Symptom Assessment

The depression status of all the subjects was quantified using the 17-item Hamilton Depression Scale (HAMD-17). The suicide item 3 was regarded as the rating for suicidal ideation (SI), which has been previously reported as an effective measure for suicidal ideation [41]. SI ratings ranged from 0 to 4, with scores ≥ 3 defined as a relatively high degree of SI (HSI) and scores < 3 as a relatively low degree of SI (LSI) [42]. Moreover, the HAMD-17 was divided into five structural factors (anxiety/somatization, cognitive disorders, psychomotor retardation, and sleep and weight disorders), which was performed to better assess the depression symptoms.

In addition, we used the Beck Scale for Suicide Ideation (BSS) to measure the current intensity of the patients’ specific attitudes, behaviors, and plans that were connected to suicidal behavior for the past week [43]. Internal reliability, test-retest stability and validity for the BSS have been established [44]. The severity of suicidal ideation was evaluated by summing the ratings of the first 19 items. Items 20 (prior suicide attempts) and 21 (severity of the suicide attempt) were not included in the total score. The sum score of the BSS ranged from 0 to 38 points. In our study, the Chinese version of the BSS was used.

The NEO-Five Factor Inventory (NEO-FFI) was used to assess personality characteristics, which is a shortened version of the NEO Personality Inventory-Revised [45]. It consists of 60 items and measures the main Big Five domains: Neuroticism (easily upset, maladjusted), Extraversion (energetic, assertive, talkative), Conscientiousness (responsible, dependable, orderly), Agreeableness (good-natured, cooperative, trusting) and Openness (imaginative, independent-minded, intellectual) [45,46]. The statement of each subject is rated based on a 5-point Likert scale, ranging from “strongly disagree” to “strongly agree”. Sum scores on each separate domain range from 12 to 60.

### 2.3. Measurements

All the subjects underwent a peripheral venous blood draw between 08:00 and 10:00 am. The samples were collected in anticoagulant-free tubes and immediately centrifuged for 20 min at a speed of 3000 r/min to obtain patients’ plasma. Plasma samples were then separated and stored at −80 °C until future analysis. ELISA kits were used to measure plasma levels. The details of all the ELISA kits were as follows: mBDNF (DBD00—R&D Systems, Minneapolis, MN, USA); tPA (DTPA00—R&D Systems, Minneapolis, MN, USA); and IL-1β (ab214025—Abcam, Cambridge, MA, USA). Plasma proBDNF, TrkB, TNF-α, and IL-6 levels were estimated using a DuoSet human ELISA kit (proBDNF: DY3175; TrkB: DYC397; TNF-α: HSTA00E; IL-6: HS60DC—R&D Systems, Minneapolis, MN, USA) combined with a DuoSet ELISA Ancillary Reagent kit (DY008—R&D Systems, Minneapolis, MN, USA). Based on the manufacturers’ instructions, we determined the concentration of factors in each sample. The results are exhibited in pg/mL. All the experiments were performed in duplicate.

### 2.4. Data Analysis

For normally distributed continuous variables, we present the data as mean ± standard deviation, while for non-normally distributed continuous variables, data are shown as median ± interquartile range. Comparisons between two groups were conducted using Students’s *t*-tests or the Mann–Whitney U test according to variance analysis. Comparisons among three groups were first conducted with the Kruskal–Wallis (K-W) test. If a significant difference was obtained, a comparison between the two groups was then carried out using the Mann–Whitney U test. Correlation analysis was performed to discover the relationship between the plasma levels of the factors and the clinical characteristics. All statistical analyses included in the study were carried out using the Statistical Package for the Social Sciences (SPSS) version 22.0 with significance at a *p* value < 0.05.

### 2.5. Ethics

This study was carried out in compliance with the Code of Ethics of the World Medical Association (Declaration of Helsinki) for experiments involving humans. Our work was approved by the Research Ethics Committee of the First Affiliated Hospital, College of Medicine, Zhejiang University (No. 2016029), and we obtained written informed consent from each subject or their family members (for the severe depression study) before their involvement in the study.

## 3. Results

### 3.1. Demographic and Clinical Characteristics

We divided all the subjects into three groups, namely healthy controls (CON), MDD with lower suicidal ideation (MDD + LSI), and MDD with higher suicidal ideation (MDD + HSI). All the demographic and clinical characteristics of the recruited subjects are exhibited in Table 1. There were no significant between-group differences regarding age, sex, and years of education or significant differences in illness duration, relapse time, and onset age between groups with high or low suicidal ideation.

HAMD-17 scores including the five structural factors were significantly different between the MDD and CON groups, with both MDD groups scoring higher than the CON groups. The scores for the “anxiety/somatization” and “cognitive disorder” structural factors were higher in the MDD + HSI group than the MDD + LSI group (*p* = 0.026 and *p* < 0.001). Higher neuroticism and lower extraversion, conscientiousness, agreeableness, and openness were observed in both MDD groups compared to the healthy controls (all *p* < 0.023). Further, lower openness was found in the MDD + HSI group compared with the MDD + LSI group (*p* = 0.024).

### 3.2. Increased mBDNF in MDD Patients with High and Low Suicidal Ideation

Table 3 shows the changes in different plasma peptides for the three groups. The levels of mBDNF, tPA, IL-1β, and IL-6 were significantly elevated in all MDD subjects compared to the healthy controls (Table 3, Figure 1). Meanwhile, the levels of TrkB and proBDNF were lower in the MDD subjects. There was no statistically significant difference in TNF-α levels among the three groups (Table 3, Figure 1). Further analysis revealed a significant difference in serum mBDNF levels between the MDD + HSI and MDD + LSI groups (8900.82 ± 5488.47 vs. 6157.71 ± 2965.86 pg/mL, respectively, *p* = 0.010, Figure 1). Moreover, we found a positive link between mBDNF and the Beck Scale sum score for MDD patients (rho = 0.299, *p* = 0.006, Figure 2).

## 4. Discussion

In the present study, we found that the mBDNF level was correlated with the level of suicidal ideation in MDD patients. For unmedicated subjects with MDD, the mBDNF levels were remarkably greater in those who had higher suicidal ideation than those with lower suicidal ideation. In MDD patients, the mBDNF level in the peripheral plasma was positively correlated with the Beck Scale scum score for suicide ideation. In addition, compared with control subjects, the plasma levels of mBDNF, tPA, and IL-1β were higher in MDD patients. However, the levels of TrkB and proBDNF were decreased in MDD patients.

Depression, which has high heterogeneity, features various clinical symptoms and drug sensitivities and could increase suicide risk independently. The lifetime prevalence of suicide in MDD patients is about 2–12% [47]. This shows that MDD is accompanied by a 20-fold elevated risk of suicide [48]. In addition, 15% of patients with MDD in the community have reported a suicide attempt at least once in their lifetime [49]. However, the etiology between suicide and MDD remains elusive and complex. Many risk factors have been regarded as stimulators of the risk of suicidal behavior (attempt or ideation), such as biological, psychological, and environmental factors.

Many studies have demonstrated that BDNF is implicated in the pathophysiology of MDD, and it is one of the main biomarkers proposed in association studies with suicide [50,51,52]. A recent study identified that the BNDF signaling pathway participates in resilience to stress [53]. Therefore, BDNF is a vital link in the pathogenesis of depression. In this study, the levels of plasma biological factors related to BDNF metabolism were analyzed, including mBDNF, proBDNF, TrkB, and tpA, in MDD patients with relatively high suicidal ideation compared to those with relatively low suicidal ideation and healthy controls. Previous studies have mainly focused on plasma/serum BDNF levels, but the results have differed. Several studies have shown that lower levels of BDNF are linked with a higher risk of suicide attempts and ideation or suicide behavior in MDD patients [54,55,56]. A recent meta-analysis showed that plasma but not serum BDNF levels were decreased in patients who had attempted suicide [57]. There were some mixed results reporting no significant association between BDNF and suicide in MDD individuals [58,59,60]. However, in the present study, we found that high levels of mBDNF were related to high suicidal ideation. It has been reported that BDNF may undergo some mutation or methylation in MDD, which influences its expression and effects in real conditions [61,62]. Thus, the mutation of BDNF in MDD may be one of the reasons for these differences. Another reason for our study’s data being inconsistent with those of most of the studies mentioned above may lie in the sample size and the standard of classification. Furthermore, our sample was composed of inpatients from a clinical ward whose suicidal ideations were evaluated immediately after drawing blood. Finally, the BDNF metabolism-related factors were measured near to the time of suicidal ideation and could be represented more effectively. In other studies, researchers have included patients with a lifetime history of suicide attempts, though BDNF measurements could not represent the levels at the time of the attempt [63]. Thus, the relationship between BDNF and suicide in MDD patients still requires deeper exploration.

However, some limitations existed in our study. First of all, our sample size was relatively small and thus could have limited statistical power to some extent. Second, the level of peripheral factors we measured could not exactly replicate the exact level of the factors in the brain. Taking BDNF as an example, peripheral BDNF has an indefinite relationship with BDNF levels in the central nervous system [55,57]. Third, other potential variables that could influence the levels of biomarkers such as exercise, genetic polymorphism, and smoking habits were not considered in this study, which could have caused bias [15,64,65]. In the future, repeating our study with a larger sample size and more detailed information on the participants would strength the findings. In addition, our study was a cross-sectional study, and therefore it was very difficult to indicate whether mBDNF associations with suicidal ideation were “state” or “trait” markers in the research sample.

In summary, this study first found a correlation between mBDNF levels and suicidal ideation. At the same time, the differences between mBDNF and proBDNF levels and their correlation with suicidal ideation in MDD patients were verified and analyzed, suggesting that serum BDNF levels may be used as a biomarker reflecting suicidal ideation in MDD patients to a certain extent. In addition, our findings also confirmed the presence of BDNF signaling-pathway-related factors (tPA, TrkB) and several major immune factor (IL-1β, IL-6, and TNF-α) abnormalities in the peripheral blood of MDD patients.

## 5. Conclusions

Overall, our data suggest that mature BDNF might be involved in the biological mechanisms of suicide ideation in MDD. Compared with healthy controls, the plasma levels of mBDNF, tPA, and IL-1β were higher in MDD patients, which indicates that inflammation is involved in the pathogenesis of depression. However, the levels of TrkB and proBDNF were decreased in MDD patients, indicating that multiple signaling pathways are implicated in MDD, with a complex pathology. In short, our study provides novel insights regarding the relationship between BDNF-TrkB signaling, inflammatory factors, and clinical characteristics in MDD, especially the possible role of mBDNF in the neurobiology of the connection between depression and suicidal ideation.

## Figures and Tables

**Figure 1 brainsci-13-01223-f001:**
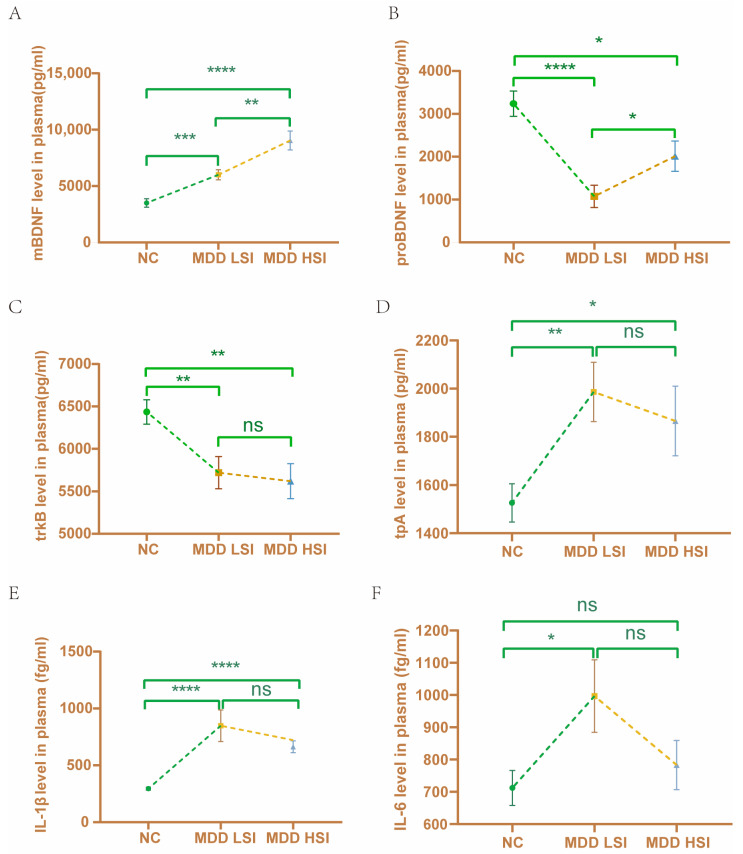
Changes in plasma peptide levels among patients with different suicidal ideation levels and healthy controls. (**A**) The levels of mBDNF were higher in MDD patients (7578.84 ± 4638.34 pg/mL) compared to controls (3508.85 ± 3590.37 pg/mL) (*** *p* < 0.001, ** *p* < 0.01, **** *p* < 0.0001), and there were significant differences between the plasma mBDNF levels in the MDD + HSI and MDD + LSI groups (8900.82 ± 5488.47 vs. 6157.71 ± 2965.86 pg/mL, * *p* = 0.010). (**B**,**C**) The levels of TrkB and proBDNF were lower in MDD subjects compared to healthy controls (5669.14 ± 1273.94 pg/mL vs. 6435.37 ± 1387.59 pg/mL, *** *p* < 0.001; 1560.51 ± 2063.48 pg/mL vs. 3236.87 ± 2832.83 pg/mL, * *p* < 0.05, ** *p* < 0.01,*** *p* < 0.001, **** *p* < 0.0001). (**D**–**F**) The levels of tPA, IL-1β, and IL-6 were significantly higher in all MDD subjects compared to healthy controls (1923.93 ± 866.90 pg/mL vs. 1526.29 ± 767.85 pg/mL, *** *p* < 0.001; 782.82 ± 655.53 fg/mL vs. 295.25 ± 126.54 fg/mL, *** *p* < 0.001; 886.00 ± 617.10 fg/mL vs. 712.08 ± 521.99 fg/mL, * *p* < 0.05, ** *p* < 0.01,*** *p* < 0.001, **** *p* < 0.0001).

**Figure 2 brainsci-13-01223-f002:**
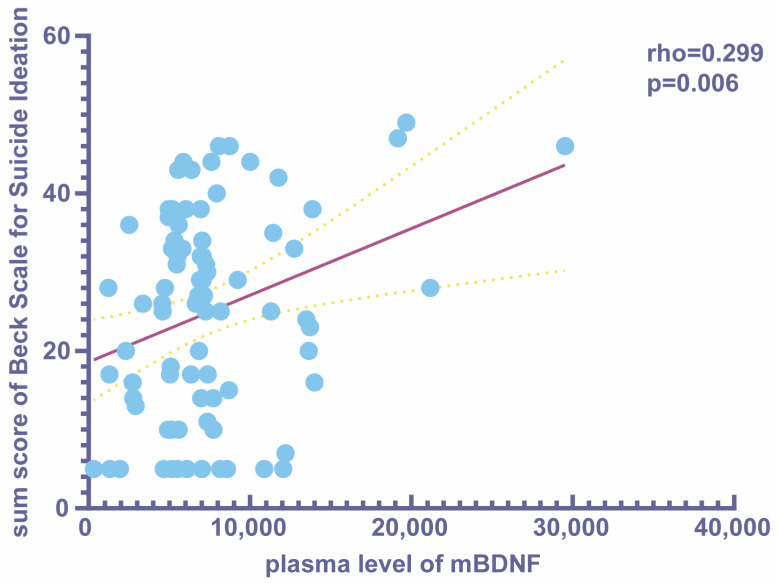
Correlations between plasma peptide levels and clinical characteristics in MDD patients: a positive correlation was found between mBDNF and the Beck Scale sum score for MDD patients.

**Table 1 brainsci-13-01223-t001:** Demographic and clinicopathological information of the included subjects.

Characteristic	MDD + HSI (*n* = 43)	MDD + LSI (*n* = 40)	CON (*n* = 93)	*p*
% Males (*n*)	37.2 (16)	35.0 (14)	43.0 (40)	0.638
Age (y), mean (SD)	26.44 (6.77)	27.98 (6.94)	27.58 (7.69)	0.566
Education (y), mean (SD)	14.21 (2.24)	14.88 (1.77)	14.92 (1.81)	0.194
Illness duration (m), mean (SD)	2.45 (2.92)	1.86 (2.05)		0.970
Relapse time (*n*)	1.53 (0.86)	1.42 (0.89)		0.321
Onset age (y)	24.35 (6.94)	25.97 (6.26)		0.272
HAMD-17 score, mean (SD)	24.53 (4.70)	23.48 (4.18)	2.28 (2.53)	<0.001 ^ab^
Anxiety/somatization, mean (SD)	8.30 (1.94)	7.26 (1.90)	1.11 (1.31)	<0.001 ^abc^
Cognitive disorder, mean (SD)	5.58 (1.43)	3.75 (1.72)	0.13 (0.37)	<0.001 ^abc^
Psychomotor retardation, mean (SD)	7.49 (1.88)	7.58 (1.78)	046 (0.93)	<0.001 ^ab^
Sleep disorder, mean (SD)	3.56 (1.59)	3.38 (1.58)	0.38 (0.90)	<0.001 ^ab^
Weight, mean (SD)	0.65 (0.90)	0.48 (0.85)	0.05 (0.31)	<0.001 ^ab^
Neuroticism, mean (SD)	41.12 (6.74)	42.10 (6.05)	27.02 (6.61)	<0.001 ^ab^
Extraversion, mean (SD)	28.23 (7.62)	30.65 (6.74)	40.11 (6.44)	<0.001 ^ab^
Conscientiousness, mean (SD)	38.14 (7.94)	38.80 (7.12)	43.56 (5.49)	<0.001 ^ab^
Agreeableness, mean (SD)	40.56 (5.84)	42.15 (5.54)	43.28 (4.62)	0.023 ^ab^
Openness, mean (SD)	35.05 (6.89)	38.03 (5.25)	39.19 (4.89)	0.001 ^abc^

Abbreviations: HAMD-17, the 17-item Hamilton Depression Scale; HSI, high suicidal ideation; LSI, low suicidal ideation; m, month; MDD, major depressive disorder; *n*, number; SD, standard deviation; y, year. ^a^ Indicates a statistically significant difference (*p* < 0.05) between the CON and MDD + HSI groups. ^b^ Indicates a statistically significant difference (*p* < 0.05) between the CON and MDD + LSI groups. ^c^ Indicates a statistically significant difference (*p* < 0.05) between the MDD + HSI and MDD + LSI groups.

**Table 2 brainsci-13-01223-t002:** Inclusion and exclusion criteria.

Inclusion Criteria	Exclusion Criteria
MDD diagnosis	Current pregnancy
Free of any psychotropic medication for at least three months	Other current psychosis or a history of bipolar disorder or psychotic symptoms
In a stable state	Eating disorder or post-traumatic stress disorder (PTSD)
	Alcohol or drug dependence or abuse
	Neurological disorder history
	MDD patients who had received any form of treatment prior to the study
	Medical illnesses that could be etiologically related to the ongoing depressive episode
	Subject refused to participate in the study

**Table 3 brainsci-13-01223-t003:** Plasma peptide levels (pg/mL) in MDD patients with high and low suicidal ideation and healthy controls.

Plasma Peptide	MDD + HSI (*n* = 43)	MDD + LSI (*n* = 40)	CON (*n* = 93)	*p*	ES (HSI vs. LSI)	ES (HSI vs. CON)	ES (LSI vs. CON)
mBDNF, mean (SD)	8900.82 (5488.47)	6157.71 (2965.86)	3508.85 (3590.37)	<0.001 ^abc^	0.30	0.50	0.37
proBDNF, mean (SD)	2011.14 (2317.79)	1076.08 (1643.96)	3236.87 (2832.83)	<0.001 ^ab^	0.23	−0.23	−0.42
TrkB, mean (SD)	5620.79 (1350.27)	5721.11 (1201.44)	6435.37 (1387.59)	<0.001 ^ab^	−0.04	−0.29	−0.27
tpA, mean (SD)	1865.83 (945.97)	1986.40 (780.20)	1526.29 (767.85)	<0.001 ^ab^	−0.07	0.19	0.28
IL-1β, mean (SD)	722.32 (341.67)	847.86 (877.17)	295.25 (125.54)	<0.001 ^ab^	−0.09	0.64	0.40
IL-6, mean (SD)	783.01 (500.53)	996.71 (711.76)	712.08 (521.99)	0.021 ^ab^	−0.17	0.07	0.22
TNF, mean (SD)	473.34 (216.04)	532.12 (196.36)	533.93 (286.75)	0.492	−0.14	-0.12	−0.00

Abbreviations: HSI, high suicidal ideation; LSI, low suicidal ideation; m, month; MDD, major depressive disorder; mBDNF, mature brain-derived neurotrophic factor; proBDNF, precursor brain-derived neurotrophic factor; SD, standard deviation; TNF, tumor necrosis factor; tpA, tissue plasminogen activator; TrkB, tropomyosin receptor kinase B. ^a^ Indicates a statistically significant difference (*p* < 0.05) between the CON and MDD + HSI groups. ^b^ Indicates a statistically significant difference (*p* < 0.05) between the CON and MDD + LSI groups. ^c^ Indicates a statistically significant difference (*p* < 0.05) between the MDD + HSI and MDD + LSI groups.

## Data Availability

The data presented in this study are available on request from the corresponding author.

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
