# Peer review of "Elevated Plasma Levels of Mature Brain-Derived Neurotrophic Factor in Major Depressive Disorder Patients with Higher Suicidal Ideation"

_brainsci, 2023, doi:10.3390/brainsci13081223_

Round 1

Reviewer 1 Report

The study is a cross-sectional investigation examining the relationship between depressive symptoms and biological markers. Overall, it presents intriguing findings that shed light on the biological background of MDD patients. However, I have identified some concerns that warrant evaluation:

  1. Hypotheses: It is recommended to include the hypotheses at the end of the introduction to provide a clear framework for the study.

  2. DSM IV criteria: Consider using more updated criteria as DSM IV is considered outdated.

  3. Clinical stability assessment: It is crucial to evaluate clinical stability before including participants in the study, as it could significantly impact the results.

  4. Role of sex: Due to the imbalance in the number of males and females, it is essential to assess the potential influence of sex on the study's results.

  5. Classification of HSI and LSI: Clarify whether the classification into High Symptom Index (HSI) and Low Symptom Index (LSI) groups was solely based on self-report questionnaires or if other methods were employed.

  6. Effect size reporting: Due to the large number of analyses, include effect size measures for significant results to better understand the practical significance of the findings.

  7. p-value correction: Change p = 0.000 to p < 0.001 in the tables to adhere to appropriate statistical reporting standards.

Reviewer 2 Report

The present article evaluates the role of increased plasma levels of mature BDNF in major depression depressive disorder patients with higher suicidal ideation. The subject is current and topical, but some deficiencies in content and form have been identified and presented below:

The abstract headings do not have to appear in the abstract but only underline each subsection of the information presented.

Brackets are preferable to the information presented using ":'" 

Directly abbreviated words should be explained at the first mention in the text, and then only the abbreviated form should be used (IL, tPA, TNF-alpha). The abstract should be treated separately from the main text in terms of abbreviations.

A comprehensive review of the English used in the article (L36 suffering instead of suffered etc.) is necessary.

World Health Organization should be written in capital letters.

L44-L69 - the paragraph is too long, thus reducing the clarity of the information. Suggest changing to smaller and clearer paragraphs.

The introduction section is poorly presented in relation to the complexity of the pathology and its management. 
It is absolutely necessary for a comprehensive introduction to present in more detail the pathophysiological mechanism of depression (including in the form of a clarifying figure) as well as some biomolecular pathways that can be targeted in the development of depression treatments. I suggest checking and referring to new and promising bibliographic sources: PMID: 34922112; PMID: 33015736; PMID: 34801611.

The scope of the paper in the last part of the introduction is recommended to be improved in terms of contribution to the scientific literature and novelty rather than presentation of what has been done.

It is advisable to present in more detail the inclusion/exclusion criteria of patients in the study, perhaps in the form of a diagram, for greater clarity.

L175- it would be advisable to put the number and date of approval of the study by the ethics committee.

The full title of the figures should not be bolded.

L237 - what is the logic of putting (1), (2) in the text?

The discussion section should be enhanced with more relevant data from the literature related to the implications of BDNF in depression. I suggest checking and referring to: PMID: 33230708; PMID: 35993165; and PMID: 31636250.

I suggest that the limitations part be included in the last part of the discussion and not as a separate part in a chapter.

I recommend that the conclusion section be less informal in terms of language. Please revise.

Reviewer 3 Report

The study explores the correlation between plasma levels of mature brain-derived neurotrophic factor (BDNF) and suicidal ideation in individuals with MDD. BDNF is a critical protein involved in neuroplasticity and neuronal survival, and its role in mental health disorders is of great interest to the scientific community.

The research methodology appears robust, and the sample size seems adequate to draw meaningful conclusions. By identifying elevated plasma levels of mature BDNF in MDD patients with higher suicidal ideation, the authors shed light on potential biomarkers that could aid in risk assessment and treatment planning for suicidal patients with depression.

However, one aspect that deserves attention is the lack of comments or discussions in the article. Peer review and constructive comments are vital to refining research and validating its findings. Although the authors' work is commendable, an absence of critical comments could leave some questions unanswered and hinder the advancement of the field.

In summary, the article makes a substantial contribution to the understanding of MDD and suicidal ideation, and the authors exhibit commendable scientific maturity. Nevertheless, future studies and comments from the scientific community would further strengthen and validate their findings.

Round 2

Reviewer 1 Report

I do not have further comments.

Reviewer 2 Report

The authors responded to my suggestions.

It should be revised as editing.